

**Air temperature changes in Wrocław (SW Poland) in 1773–81 based on a newly discovered series of meteorological measurements**

Rajmund PRZYBYLAK[1,3] (ORCID: 0000-0003-4101-6116)

Aleksandra POSPIESZYŃSKA[1,3] (0000-0003-2532-7168)

Piotr OLIŃSKI[2,3] (ORCID: 0000-0003-1428-0800)

[1] Department of Meteorology and Climatology, Faculty of Earth Sciences and Spatial Management, Nicolaus Copernicus
University, Toruń, Poland
[2] Department of Medieval History, Institute of History and Archival Sciences, Faculty of History, Nicolaus Copernicus University
in Toruń, Poland
[3] Centre for Climate Change Research, Nicolaus Copernicus University, Toruń, Poland
Corresponding authors: Aleksandra Pospieszyńska, opos@umk.pl & Rajmund Przybylak,
rp11@umk.pl
Abstract. The article presents a description of a newly discovered series of meteorological
measurements made in Wrocław (SW Poland) in 1773–81 and its importance for climate analysis. The
series is the third-oldest available for Wrocław. The 1773–81 observations were made by Johann
Ephraim Scheibel on the premises of the Gymnasium that belonged to the Church of St Elizabeth
(central Wrocław). Meteorological observations of air temperature and atmospheric pressure were
made three times a day (morning, midday and evening) – or twice a day (morning and evening) for
atmospheric precipitation, wind (including direction and force) and humidity. The meteorological
registers were published in the newspaper *Oekonomische Nachrichten der Patriotischen Gesellschaft
in Schlesien* issued in Wrocław in the years 1773–81. Between 1773 and 1776, the meteorological data
were published in each weekly edition of the newspaper. From 1777 onwards, the data were published
for the entire month and in three-month blocks in the last two years. In addition to the measurements,
J. E. Scheibel also published, in the same source, between one and three pages of weather descriptions
for each month or block of two to three months. The air temperature in Wrocław during the period
1773–81 was 1.2 °C and 0.3 °C warmer than in the respective equivalent periods 100 and 200 years
later (i.e., 1873–81 and 1973–81) but 2.1 °C colder than in the most recent period (2013–21). The
increase in temperature between the study period and the most recent period was greatest in summer
(2.8 °C) and smallest in spring (1.4 °C).

Keywords: Poland, Silesia, historical climatology, air temperature, instrumental observations, 18[th]
century
**Short summary:** The article presents: (i) a description of a newly discovered series of meteorological
measurements (air temperature, atmospheric precipitation and pressure, humidity, wind direction and
force) made in Wrocław (SW Poland) in 1773–81, and (ii) a comprehensive analysis of air temperature.
This unique series, quality-controlled and verified against the nearest series of data (Prague and
Berlin), extends the regular temperature series for Wrocław back to 1773 (it having previously begun
at 1791).





1. Introduction

Historical climatology is a dynamically evolving discipline of science, particularly in recent decades (for details, see, e.g., Brázdil et al., 2005; Brönnimann et al., 2018, 2019). Its main goal is to find, in libraries and archives, historical sources (e.g., written records such as sagas, chronicles, maps and early instrumental measurements) that contain meteorological measurements or weather descriptions and then to record them, preferably in digital form. Such activity, often referred to as "data rescue activity", significantly enhances existing meteorological databases, thereby improving knowledge about weather and climate in historical periods (i.e., before the beginning of regular observations) for many regions around the world.

In Poland, knowledge about historical climate has increased significantly since the 1990s (e.g., Sadowski, 1991; Majorowicz et al., 2004; Przybylak et al., 2005, 2010, 2020, 2023; Filipiak and Miętus, 2010; Przybylak, 2010, 2011, 2016; Przybylak and Marciniak, 2010; Hernández-Almeida et al. 2015, 2016; Filipiak et al., 2019; Ghazi et al., 2023a, b, 2024, 2025). Thorough and comprehensive reviews of the work on this topic have recently been published by Przybylak (2010, 2016), Przybylak et al. (2010) and Opała-Owczarek et al. (2021). The analysis of these works reveals that the search for early instrumental observations in Poland has spanned more than a hundred years and intensified at the beginning of the 20[th] century. From the most important published articles at this time, we should mention those of, e.g., Romer (1910), Merecki (1917), Birkenmajer and Birkenmajer (1918), Hellmann (1918), Pawłowski (1919) and Gorczyński (1922). According to Rojecki (1966), the first outline of the history of meteorological observations was presented by Baranowski (1858) in a foreword he wrote to the Polish translation of P. Foissac's book. Following World War II, Polish scientists continued the search for early meteorological data. Of the numerous available publications, the most valuable results are presented in works written by Rojecki (1956, 1966). In the first of these two articles, Rojecki's Table 1 presents a list of 30 places where observations were made before the end of the 18[th] century. In line with expectations, meteorological measurements taken in the 18[th] century were the most abundant. Note, however, that we have information that some measurements were conducted in certain places and for certain years but still have not found that data. This failure is due, for example, to not all sources in archives being precisely catalogued – or to some sources having been destroyed (or lost) due to the frequent wars that have plagued Poland's history.

The area of interest in the present paper is Wrocław, where instrumental meteorological observations began very early, i.e. at the beginning of the 18[th] century. For this period, we have two long-term series of observations: 04.1710–1721 (David von Grebner) and 1717–30 (Johann Kanold/A. E. Büchner) (see Munzar, 2003; Pyka, 2003; Przybylak, 2010; Przybylak and Pospieszyńska, 2010). For the subsequent period lasting until the end of January 1791, there was no information in any of the





above-cited literature items (old or modern) regarding the availability of meteorological observations.
Regular observations (thrice-daily) in Wrocław started in February 1791 at the Astronomical
Observatory, based in the Mathematics Tower of the Universitatis Leopoldinae Vratislaviensis (the
Latin name for Wrocław University, founded by the Jesuits) (Bryś and Bryś, 2010a, b). The series of air
temperature measurements is the second-longest series of such data in Poland (after Warsaw, where
they started in 1779).

From this brief review of the state of knowledge about the climate of the historical period in

Poland, and despite over 100 years of many Polish researchers and climatologists searching for early
instrumental meteorological observations, no one has managed to obtain information about the
existence of meteorological observations in Wrocław between 1730 and 1790. Only our archival and
library research conducted while realising the research project, *The occurrence of extreme weather,*
*climate and water phenomena in Poland from the 11th to the 18th century in the light of multiproxy*
*data,* resulted in the discovery of a new, extremely important long-term series of meteorological
observations made in the centre of Wrocław in the years 1773–81. Therefore, the primary objective of
this article is to present the newly discovered series of observations to a broader scientific audience
and to report the initial climate analysis results, albeit limited to the description of air temperature
conditions and changes in Wrocław at that time. The secondary goal is to compare air temperature in
the period 1773–81 (9 years) against more recent sets of nine-year data, including data from the
contemporary period, (1873–81, 1973–81 and 2013–21), as well as against the longer reference period
1961–90 used in the ModE-RA paleo-reanalysis (Valler et al., 2024). Additionally, a spatial coherence
of air temperature in this period was analysed using data series available for the two closest stations
(Prague and Berlin) in central Europe.

2.  Area, Data and Methods

The 1773–81 observations were made by Johann Ephraim Scheibel on the premises of St Elizabeth's
High School, which belongs to the Church of St Elizabeth (central Wrocław) (Fig. 1). Johann Ephraim
Scheibel (b. 1736, d. 31 May 1809) was a mathematician and astronomer, professor (as of 1759) and
teacher of mathematics, physics and logic at the Gymnasium at the churches of St Elizabeth
(Elisabethanum) and St Mary Magdalene in Wrocław. The meteorological registers were published in
the newspaper *Oekonomische Nachrichten der Patriotischen Gesellschaft in Schlesien* issued in
Wrocław in the years 1773–81 ([https://www.deutsche-digitalebibliothek.de/item/IPAKM4M4JIS3](https://www.deutsche-digitalebibliothek.de/item/IPAKM4M4JIS3)
[KFO7ASMQ5IKW0ZWECOODS](https://www.deutsche-digitalebibliothek.de/item/IPAKM4M4JIS3KFO7ASMQ5IKW0ZWECOODS)). In the years 1773–76, meteorological data were published in each
weekly edition of the newspaper (usually seven days of observations starting on Friday and ending on
Thursday; the newspaper was issued on Saturday). From 1777 onwards, data were published for the
entire month and, in the last two years, in three-month blocks. Besides measurements, in the same



source, J. E. Scheibel also published between one and three pages of weather descriptions for each
month or block of two or three months.

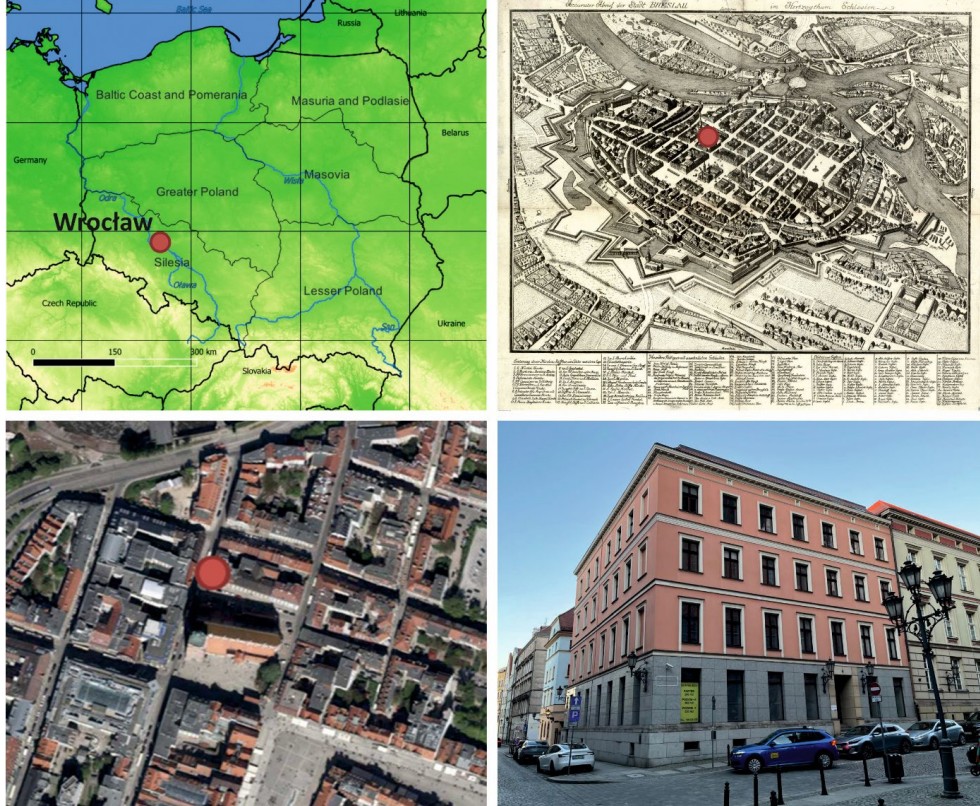


Fig. 1. Location of the site of meteorological observations conducted by Johann Ephraim Scheibel in Wrocław in the period
1773–81: (*upper left*) location of Wrocław within Poland; (*upper right*) location of the Gymnasium on a historical plan of
Wrocław from the period 1776–80 (source of map: https://polska-org.pl/7719498,foto.html?idEntity=554891, last access
13.05.2025); (*bottom left*) location of historical observation site on a contemporary orthophotomap (source of map:
https://geoportal-krajowy.pl/); (*bottom right*) St Elizabeth's High School (4, St Elizabeth St.) and the Rector's apartment where
Scheibel lived and probably made observations (photo by Rajmund Przybylak)

The newspaper *Oekonomische Nachrichten der Patriotischen Gesellschaft in Schlesien* began
publishing meteorological data on January 9, 1773 (i.e., in the very first edition of the newspaper). The
data published in this issue were for the period January 1st to 7th, 1773. The quality of this first issue
of the newspaper is poor, so we attach an example of the second issue of the newspaper (published
on January 16, 1773) (Fig. 2).



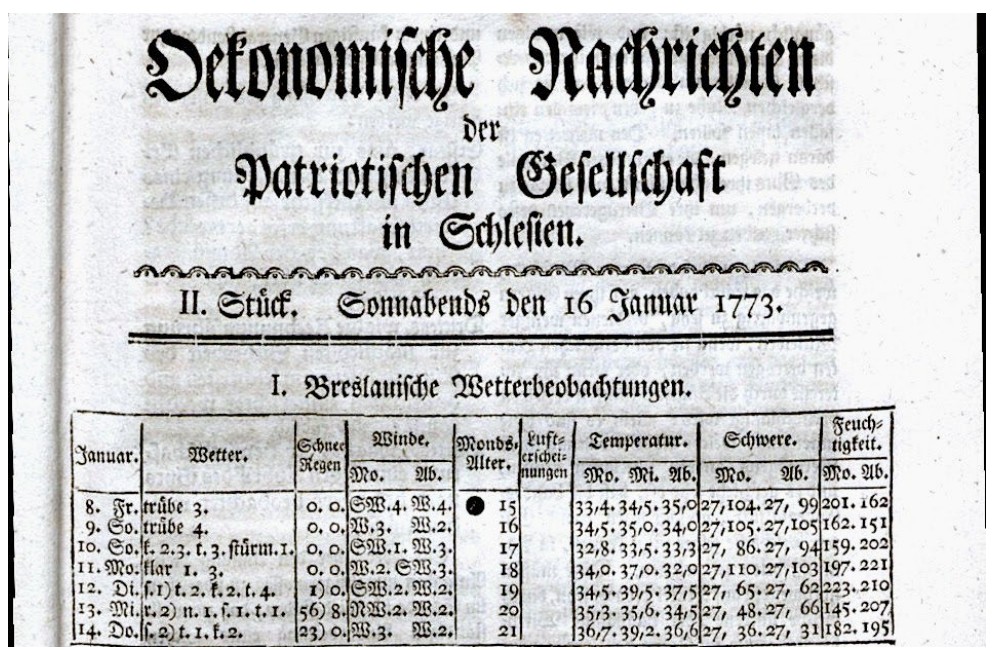

Fig. 2. Example of meteorological observations made on 8–14 January 1773 and published on page 9 of the newspaper *Oekonomische Nachrichten der Patriotischen Gesellschaft in Schlesien* on 16th January 1773

In the newspaper's first issue, published on January 9, 1773, Scheibel provides information that wind direction was estimated based on the movement of the flag on the tower of St Elizabeth's Church, meaning the tower was observable from the meteorological measurement site. Our visit to the city centre of Wrocław, the site where Scheibel made his observations, confirms this conclusion. From the windows of the building that once housed St Elizabeth's High School, we can see the tower of St Elizabeth's Church with a little flag on top, which allows us to judge the wind direction (Fig. 3).



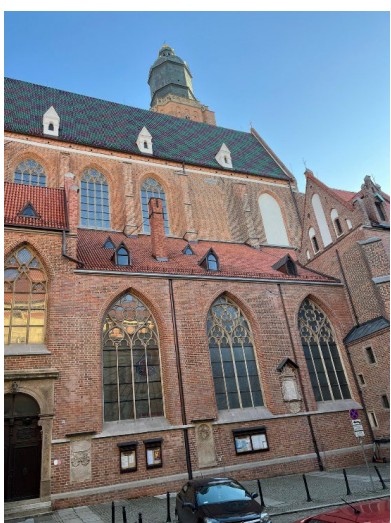


Fig. 3. St Elizabeth's Church with a little flag on the top of the tower, which allowed Scheibel to estimate wind
direction. View from the side of the former St Elizabeth's High School building (Photo by Rajmund Przybylak)

No further details about the exposition or precise location of the instrument installation are

available. Meteorological observations of air temperature (in °F) and atmospheric pressure (in French
inches) were made three times a day – in the morning, midday and evening (see Fig. 2). The other
variables were measured twice a day: atmospheric precipitation (in French inches, stating the
category: rain or snow), wind (direction [8 directions] and force [a 7-degree scale]) and humidity (using
a hygrometer produced by Prof. Johann Heinrich Lambert). Atmospheric phenomena and weather
were also described (Fig. 2). It should be noted that, to our knowledge, these air humidity
measurements are the first to have been conducted in Poland.

In this paper, we present air temperature conditions in Wrocław from 1773 to 1781, based on

both sub-daily data (morning, midday and evening) and mean daily values calculated using the
weighted mean: (Tmorning + Tmidday + 2 × Tevening)/4. The lack of information on the exact times of
observations in historical periods may introduce some inaccuracy when comparing their results to
averages calculated for the later nine-year periods (1873–81, 1973–81 and 2013–21). For the first
period, the exact observation times (morning, midday and evening) are also unknown, whereas, for
the other two periods, we calculated daily means using weighted averages based on data from 6, 12
and 21 hours. All original historical measurements from the late 18[th] century were first quality-
controlled by visual inspection and then converted to the thermometric scale currently in use, i.e.,
degrees Celsius. The data, thus processed, are available in the repository of Nicolaus Copernicus
University: https://doi.org/10.18150/PYVVWU. Three series of raw data were taken from this database
to calculate standard statistics describing climate (daily, monthly, seasonal and annual means, and



standard deviations based on sub-daily and daily data; boxplots and frequencies of occurrence of
temperatures stratified into 1-degree intervals). The historical data were compared against newer data
of similar duration (from the 19th, 20th and 21st centuries) taken from different times of the day,
representing morning, midday and evening hours. For the purposes of comparison against other
neighbouring series (Berlin and Prague) and against data taken for Wrocław but from the ModE-RA
paleo-reanalysis (Valler et al., 2024), the series of daily weighted mean temperatures were corrected
to real daily temperatures calculated using data from 24 hours. For this purpose, contemporary hourly
data from Wrocław for the period 2013–21 have been used. The values of introduced corrections for
the particular months are shown in Table S1. If the historical observations were made within the times
06:00–07:00, 12:00–13:00 and 20:00–21:00 (formulas 7–10), then the biases are very small and on
average do not exceed ±0.2 °C (see Table S1). On the other hand, maximum biases range from 0.1 °C
(December and January) to -0.5 °C (August to October). The correction of historical mean monthly
temperature data in Wrocław was necessary because the Prague series was homogenised, which
means that the historical data were also corrected to be comparable to present-day data. The Berlin
series was probably not homogenised for the 18[th] century until the mid-19th century, due to the
various relocations of stations and changes in instruments without proper documentation (see Cubash
and Kadow, 2011).

Note that the research results presented for Wrocław for the historical period contain

uncertainties that are difficult to determine due to the lack of information on the thermometer's
exposure, the type of thermometer used, and the exact times at which readings were taken, among
other factors. The thermometer, however, was probably placed on the north-west side of the High
School building, where Scheibel lived as the Rector of the school (Fig. 1).

3.  Results

The newly discovered series of meteorological measurements for Wrocław for the historical period
1773–81 is quite long (9 years). As we mentioned, this is the third-oldest time series of weather
observations for this place, and, more importantly, the new data partly fill the data gap in Wrocław
that, prior to this study, covered the period 1731–90. The series of data described here allows for an
approximate characterisation of climate conditions at this time. In this paper, we initiate an analysis of
climate conditions, focusing primarily on air temperature, which is the most significant variable of
climate in moderate latitudes, including Poland.

3.1. Monthly resolution

The annual cycle of air temperature changes in Wrocław in the studied historical period is presented
using monthly means calculated for all three sub-daily measurement times and using average daily



values (Table 1 and Table S2, Fig. 4). On average, in line with expectations, the warmest month was
July (19.3 °C) and the coldest was January (-2.5 °C). Only slightly colder than July was August (19.2 °C).
On the other hand, the other two winter months (December and February) were much warmer than
January, with even positive temperatures of 1.3 and 0.9 °C, respectively (Table S2, Fig. 4). The warmest
summers occurred in 1775 and 1781 with average temperatures of 20.6 °C and 20.5 °C, respectively
(Table 1, Fig. 5), whereas the coldest occurred in 1777 (17.6 °C). Winters were warmest and coldest in,
respectively, 1779 (2.2 °C) and 1780 (-2.5 °C). On average, the winter temperature was only slightly
below freezing (-0.2 °C), whereas the mean summer temperature reached 18.8 °C. The annual
temperature ranged between 8.3 °C (1777) and 10.6 °C (1773), with a nine-year average of 9.4 °C (Table
1 and Table S2, Fig. 5).
Table 1. Monthly, seasonal and annual mean air temperature (°C) in Wrocław for three measurement times (morning,
midday and evening) and weighted daily mean, 1773–81

| Period | J | F | M | A | M | J | J | A | S | O | N | D | DJF | MAM | JJA | SON | YEAR |
|--------|---|---|---|---|---|---|---|---|---|---|---|---|-----|-----|-----|-----|------|
| MORNING | | | | | | | | | | | | | | | | | |
| 1773 | 1.0 | -4.2 | 0.8 | 8.3 | 14.4 | 16.6 | 17.3 | 17.6 | 14.6 | 11.1 | 4.1 | 4.3 | 0.4 | 7.8 | 17.2 | 9.9 | 8.8 |
| 1774 | -1.8 | 0.7 | 3.6 | 10.3 | 13.4 | 16.3 | 16.7 | 15.2 | 10.7 | 7.3 | -2.5 | -2.9 | -1.3 | 9.1 | 16 | 5.2 | 7.2 |
| 1775 | -2.6 | 3.0 | 4.1 | 4.6 | 10.0 | 16.6 | 18.2 | 18.5 | 14.0 | 9.2 | 3.8 | 0.2 | 0.2 | 6.2 | 17.8 | 9.0 | 8.3 |
| 1776 | -10.4 | 2.4 | 3.6 | 6.0 | 9.5 | 15.6 | 17.8 | 15.8 | 12.7 | 6.8 | 3.1 | -1.0 | -3.0 | 6.4 | 16.4 | 7.5 | 6.8 |
| 1777 | -3.9 | -2.5 | 1.5 | 4.7 | 12.3 | 15.0 | 16.1 | 16.2 | 10.4 | 6.8 | 4.7 | 0.7 | -1.9 | 6.2 | 15.8 | 7.3 | 6.8 |
| 1778 | -1.9 | -2.6 | 2.7 | 8.3 | 12.1 | 15.8 | 18.1 | 16.2 | 10.6 | 6.2 | 4.3 | 4.1 | -0.2 | 7.7 | 16.7 | 7.0 | 7.8 |
| 1779 | -3.3 | 3.3 | 3.9 | 8.4 | 12.8 | 14.2 | 15.9 | 17.5 | 13.7 | 10.4 | 4.4 | 3.5 | 1.2 | 8.3 | 15.9 | 9.5 | 8.7 |
| 1780 | -4.2 | -4.0 | 4.9 | 5.6 | 11.6 | 14.2 | 16.5 | 15.4 | 11.2 | 9.0 | 3.6 | -2.3 | -3.5 | 7.4 | 15.4 | 7.9 | 6.8 |
| 1781 | -3.5 | -0.6 | 3.0 | 7.4 | 11.8 | 17.6 | 17.3 | 18.8 | 14.8 | 5.5 | 4.1 | -0.1 | -1.4 | 7.4 | 17.9 | 8.1 | 8.0 |
| MIDDAY | | | | | | | | | | | | | | | | | |
| 1773 | 2.2 | -0.8 | 4.3 | 12.6 | 21.4 | 22.7 | 23.3 | 22.8 | 20.0 | 15.4 | 6.2 | 5.6 | 2.4 | 12.7 | 22.9 | 13.9 | 13.0 |
| 1774 | -0.2 | 3.6 | 8.4 | 15.9 | 18.9 | 22.7 | 24.1 | 22.5 | 15.2 | 11.1 | 0.4 | -0.6 | 1.0 | 14.4 | 23.1 | 8.9 | 11.8 |
| 1775 | -0.3 | 5.7 | 8.7 | 9.8 | 16.5 | 24.8 | 24.8 | 24.7 | 20.8 | 13.5 | 6.1 | 2.2 | 2.5 | 11.6 | 24.8 | 13.5 | 13.1 |
| 1776 | -7.5 | 6.2 | 8.0 | 11.8 | 15.1 | 21.1 | 24.1 | 22.2 | 17.6 | 11.4 | 6.1 | 1.4 | 0.0 | 11.6 | 22.5 | 11.7 | 11.5 |
| 1777 | -0.7 | 0.9 | 5.1 | 9.4 | 17.9 | 21.1 | 21.0 | 22.0 | 15.4 | 12.0 | 7.3 | 2.3 | 0.8 | 10.8 | 21.4 | 11.5 | 11.1 |
| 1778 | 0.6 | 1.0 | 7.3 | 15.0 | 18.1 | 21.3 | 24.8 | 21.5 | 14.6 | 10.2 | 7.3 | 5.5 | 2.4 | 13.5 | 22.5 | 10.7 | 12.3 |
| 1779 | -0.7 | 7.6 | 9.8 | 15.9 | 19.2 | 19.5 | 22.1 | 23.1 | 19.2 | 14.7 | 6.8 | 4.9 | 3.9 | 15.0 | 21.6 | 13.6 | 13.5 |
| 1780 | -1.4 | -0.5 | 9.9 | 11.0 | 17.2 | 19.7 | 22.6 | 22.7 | 17.6 | 13.5 | 5.6 | -0.6 | -0.8 | 12.7 | 21.7 | 12.2 | 11.4 |
| 1781 | -1.2 | 2.5 | 6.6 | 14.1 | 18.6 | 24.5 | 24.2 | 26.4 | 20.8 | 10.3 | 6.9 | 2.1 | 1.1 | 13.1 | 25.0 | 12.7 | 13.0 |
| EVENING | | | | | | | | | | | | | | | | | |
| 1773 | 1.1 | -2.4 | 2.0 | 9.8 | 17.1 | 18.5 | 19.0 | 19.4 | 16.3 | 12.2 | 4.6 | 4.4 | 1.0 | 9.7 | 19.0 | 11.0 | 10.2 |
| 1774 | -2.0 | 1.2 | 4.8 | 11.9 | 15.1 | 18.4 | 18.7 | 16.8 | 11.3 | 7.7 | -1.9 | -2.6 | -1.1 | 10.6 | 18.0 | 5.7 | 8.3 |
| 1775 | -2.2 | 3.3 | 4.9 | 5.5 | 11.5 | 19.1 | 19.7 | 20.6 | 16.8 | 9.9 | 4.8 | 0.6 | 0.6 | 7.3 | 19.8 | 10.5 | 9.5 |
| 1776 | -9.7 | 3.9 | 4.5 | 7.6 | 10.5 | 16.8 | 19.2 | 17.0 | 13.1 | 7.1 | 3.7 | -0.6 | -2.1 | 7.5 | 17.7 | 7.9 | 7.8 |
| 1777 | -3.4 | -1.3 | 2.2 | 5.2 | 13.5 | 16.2 | 16.2 | 17.0 | 11.1 | 7.9 | 5.4 | 0.7 | -1.3 | 7.0 | 16.5 | 8.1 | 7.6 |
| 1778 | -0.8 | -1.8 | 3.4 | 9.7 | 12.9 | 16.6 | 19.3 | 17.3 | 11.4 | 7.3 | 4.7 | 4.0 | 0.5 | 8.7 | 17.7 | 7.8 | 8.7 |
| 1779 | -2.4 | 4.3 | 5.4 | 9.9 | 14.4 | 15.3 | 17.1 | 19.1 | 15.0 | 11.5 | 5.4 | 3.3 | 1.8 | 9.9 | 17.2 | 10.6 | 9.9 |
| 1780 | -3.5 | -3.3 | 5.8 | 6.7 | 13.4 | 15.4 | 17.6 | 17.4 | 13.0 | 10.0 | 4.3 | -1.6 | -2.8 | 8.6 | 16.8 | 9.1 | 7.9 |



| 1781 | -2.6 | 0.3 | 3.4 | 9.8 | 13.6 | 18.6 | 18.8 | 21.4 | 16.4 | 6.9 | 4.9 | 0.0 | -0.8 | 8.9 | 19.6 | 9.4 | 9.3 |
|---|---|---|---|---|---|---|---|---|---|---|---|---|---|---|---|---|---|
| MEAN DAILY | | | | | | | | | | | | | | | | | |
| 1773 | 1.4 | -2.5 | 2.3 | 10.1 | 17.5 | 19.1 | 19.7 | 19.8 | 16.8 | 12.7 | 4.9 | 4.7 | 1.2 | 10.0 | 19.5 | 11.5 | 10.6 |
| 1774 | -1.5 | 1.7 | 5.4 | 12.5 | 15.6 | 19.0 | 19.6 | 17.8 | 12.1 | 8.5 | -1.5 | -2.2 | -0.6 | 11.2 | 18.8 | 6.4 | 8.9 |
| 1775 | -1.8 | 3.8 | 5.7 | 6.4 | 12.4 | 19.9 | 20.6 | 21.1 | 17.1 | 10.6 | 4.9 | 0.9 | 1.0 | 8.1 | 20.6 | 10.9 | 10.1 |
| 1776 | -9.3 | 4.1 | 5.2 | 8.3 | 11.4 | 17.6 | 20.1 | 18.0 | 14.1 | 8.1 | 4.2 | -0.2 | -1.8 | 8.3 | 18.6 | 8.8 | 8.5 |
| 1777 | -2.9 | -1.1 | 2.8 | 6.1 | 14.3 | 17.1 | 17.4 | 18.1 | 12.0 | 8.7 | 5.7 | 1.1 | -0.9 | 7.8 | 17.6 | 8.8 | 8.3 |
| 1778 | -0.7 | -1.3 | 4.2 | 10.7 | 14.0 | 17.6 | 20.4 | 18.1 | 12.0 | 7.8 | 5.3 | 4.4 | 0.8 | 9.7 | 18.7 | 8.3 | 9.4 |
| 1779 | -2.2 | 4.9 | 6.1 | 11.0 | 15.2 | 16.1 | 18.1 | 19.7 | 15.7 | 12.0 | 5.5 | 3.8 | 2.2 | 10.8 | 18.0 | 11.1 | 10.5 |
| 1780 | -3.2 | -2.8 | 6.6 | 7.5 | 13.9 | 16.2 | 18.6 | 18.2 | 13.7 | 10.6 | 4.5 | -1.5 | -2.5 | 9.3 | 17.7 | 9.6 | 8.5 |
| 1781 | -2.5 | 0.6 | 4.1 | 10.3 | 14.4 | 19.8 | 19.8 | 22.0 | 17.1 | 7.4 | 5.2 | 0.5 | -0.5 | 9.6 | 20.5 | 9.9 | 9.9 |



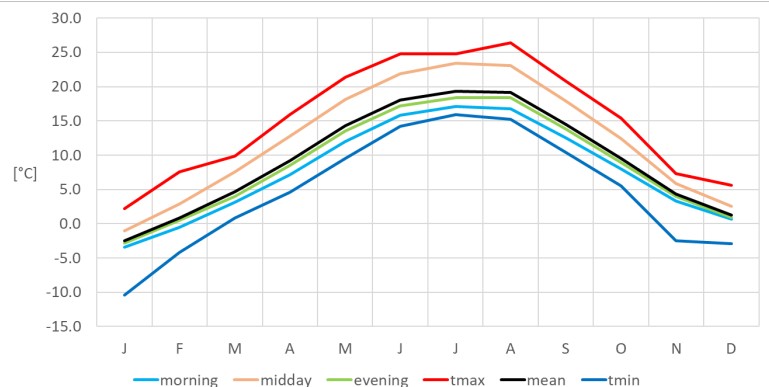


Fig. 4. Annual cycle of monthly means of air temperature (°C) in Wrocław for three measurement times (morning, midday and evening) and for daily means (mean), 1773–81. Key: tmax – highest monthly, seasonal and annual value extracted from three measurement times; Tmin – lowest monthly, seasonal and annual value extracted from three measurement times


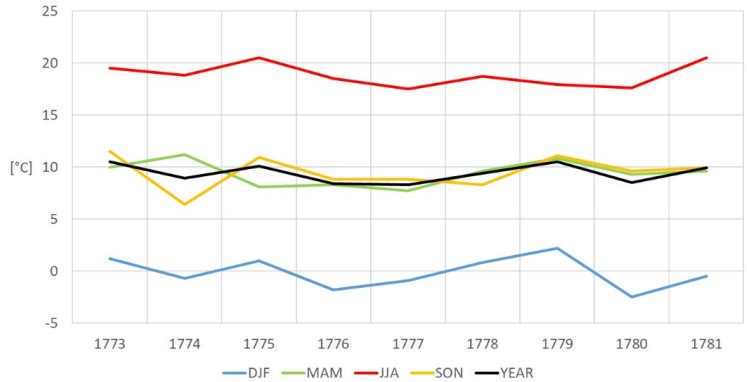


Fig. 5. Year-to-year changes in seasonal and annual mean air temperature in Wrocław, 1773–81






Table S2 and Fig. 4 present the annual cycles of air temperature at three sub-daily
measurement times, along with their highest and lowest mean monthly values. The courses of curves
are generally similar to the mean annual temperature cycle described earlier. The average annual air
temperatures at specific measurement times (morning, midday and evening) were 7.7 °C, 12.3 °C and
8.8 °C, respectively. The annual cycle based on mean monthly temperature values is clearly better
approximated by the temperature observed in evening hours, particularly in the cold half-year, when
the differences are about 0.4 °C (see Table S2, Fig. 4). For all measurement times, the highest
temperature was recorded in July and the lowest in January. The air temperature was greater in
evening hours than morning hours, especially in the warm half-year (see Table S2 and Fig. 4). The
annual range between the highest and lowest temperatures was clearly greatest for midday
measurements (24.4 °C) and smallest for morning measurements (20.5 °C). The absolute range
between mean monthly values measured at midday and morning hours reached 36.8 °C. The monthly
mean temperature was highest (26.4 °C) for August 1781 (midday) and lowest (-10.4 °C) for January
1776 (morning) (Table 1). On the other hand, the highest single temperature measurement (33.9 °C)
was recorded for midday of July 4, 1781 and the lowest (-22.8 °C) in the morning of January 27, 1776.
The preliminary analysis of the values revealed that data from midday can be treated roughly as the
maximum observed value during the day. On the other hand, the morning observation seems to
represent approximately the minimum temperature value for the day.

3.2. Daily and sub-daily resolution
The courses of curves representing the different times of day (morning, midday and evening) and the
course of daily means show very good correspondence in Wrocław during the study period (Fig. 6).
Midday temperatures were markedly higher than morning and evening values on almost all days, with
the size of the positive temperature differences being especially large during the summer months. It
cannot be ruled out that these large differences, primarily resulting from changes in the Sun's height
during the day, can also be somewhat related to the thermometer having been insufficiently protected
against solar radiation (Böhm et al., 2010). Both of these reasons are less important in winter, and, as
a result, the smallest temperature differences occur during this season.
The highest air temperatures in Wrocław during the study period were recorded from
approximately the 5th to the 15th of August (daily mean > 20°C), while the coldest temperatures were
clearly noted in the second half of January (from about -2 to -4°C). Furthermore, the temperature is,
on most days, lowest in the morning, whereas the evening temperature most closely approximates the
daily mean (Fig. 6).



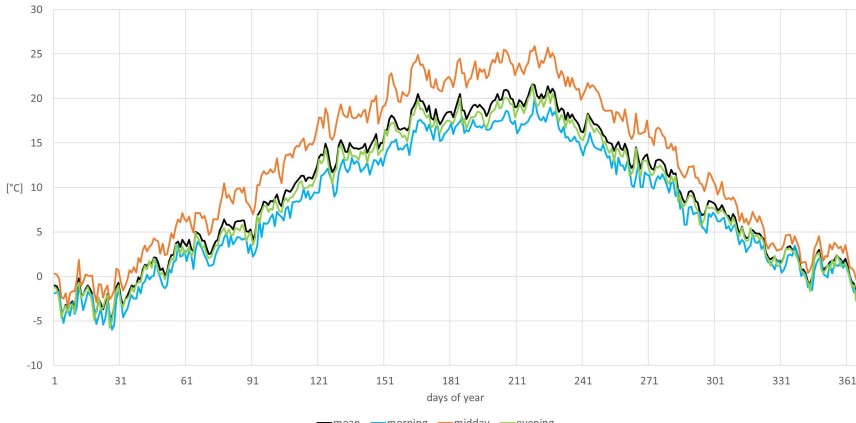


Fig. 6. Annual cycle of average air temperature in Wrocław based on sub-daily (morning, midday and evening) and daily means
(mean), 1773–81

The range of change in the annual cycle of the average temperature at noon (midday) varied in the study period from below 0 °C in January to above 20 °C, and on some days even >25 °C in July and (especially) August (Fig. 6). Morning air temperatures exceeded 15 °C (but not 20 °C) in summer, whereas, in winter, they usually ranged between 0 °C and -5 °C. Thermal winter (mean daily temperature <0 °C) extended only from the end of December to about 10th February (i.e., less than 50 days). On the other hand, on average, thermal summer (mean daily temperature >15 °C) started on 1$^{st}$ June and ended on about 10 September (i.e., summer was about twice as long as winter) (Fig. 6).

More details about the distributions of all the studied series of air temperatures are presented in Fig. 7. Discounting absolute values, all months are roughly similar to one another in terms of the range and distribution of groups of analysed temperature data. This similarity holds even when including the distributions of outliers (the dots in the boxplots). In all series, the majority of outliers occurred from November to January. It is also worth noting that cold outliers are significantly more numerous than warm ones. In most cases, the monthly median values are centrally located in the boxes, and the lengths of the whiskers are similar, except in the winter months (Fig. 7).





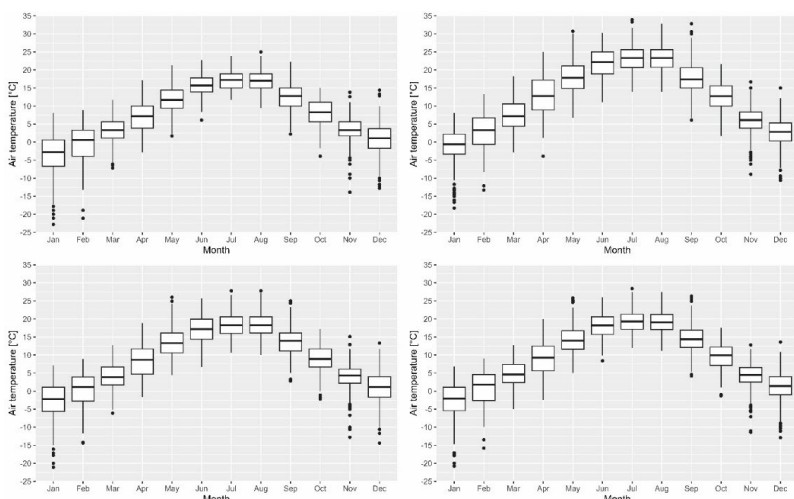


Fig. 7. Boxplots of the distribution of morning (upper left), midday (upper right), evening (lower left) and mean daily (lower
right) air temperatures in Wrocław, 1773–81

The extent to which the individual daily values measured in three measurement times

(morning, midday and evening) and mean daily values differ from their monthly mean values was
assessed in terms of the standard deviation (SD, Table 2). In line with expectations, the day-to-day
variability in daily mean temperatures is smaller than the analogous changes in temperature between
the three measurement times studied.
Table 2. Mean monthly, seasonal and annual standard deviations (SD) of air temperature (°C) in Wrocław for three
measurement times (morning, midday and evening) and daily mean, 1773–81

| Period | Jan | Feb | Mar | Apr | May | Jun | Jul | Aug | Sep | Oct | Nov | Dec | DJF | MAM | JJA | SON | Year |
|---|---|---|---|---|---|---|---|---|---|---|---|---|---|---|---|---|---|
| Tmorning | 4.5 | 4.0 | 3.4 | 3.6 | 3.3 | 2.7 | 2.4 | 2.8 | 3.0 | 3.3 | 3.3 | 3.8 | 4.1 | 3.4 | 2.6 | 3.2 | 3.4 |
| Tmidday | 4.0 | 3.9 | 3.8 | 4.6 | 4.5 | 3.9 | 3.5 | 3.3 | 3.5 | 3.5 | 3.3 | 3.4 | 3.8 | 4.3 | 3.5 | 3.4 | 3.7 |
| Tevening | 4.3 | 3.6 | 3.3 | 3.8 | 3.8 | 3.0 | 3.0 | 3.0 | 3.2 | 3.3 | 3.2 | 3.7 | 3.9 | 3.6 | 3.0 | 3.2 | 3.5 |
| Tdaily mean | 3.4 | 3.5 | 3.5 | 2.8 | 4.1 | 3.3 | 3.0 | 3.0 | 3.0 | 3.0 | 2.8 | 3.5 | 3.5 | 3.4 | 3.1 | 2.9 | 3.3 |


On average, the highest SDs (greatest variability) occurred in winter (3.5 °C) and the lowest in

autumn (2.9 °C). Surprisingly, day-to-day stability was greatest in April and November (2.8 °C), as
compared to the summer months (3.0 °C). In the daily cycle, mean annual day-to-day variability was
greatest at midday (3.7 °C) and smallest in the morning (3.4 °C). In the annual cycle, variability is
greatest at midday from March to October, whereas, in the remaining months, it is greatest in the
morning (Table 2). Finally, we should mention the great instability of day-to-day temperatures in April
and May, which, although seen in the evening, is particularly evident for midday observation hours.

At all times of observations, the range of air temperature was greater in winter and autumn

(35–40 °C) than in spring and markedly greater than in summer (when it was about 25–30 °C). One



exception here is the midday spring temperatures, which ranged by approximately 35 °C (Fig. 8). The
shapes of the temperature distributions for the transitional seasons are very similar to one another,
particularly in the morning and evening. In summer, the predominant temperature values, grouped in
intervals of 1 degree, are those from 15 to 20 °C during morning and evening observation times,
whereas for midday they range from 22 to 26 °C. In line with expectations, the distribution of mean
daily temperatures is more regular and closer-to-normal than the distribution of morning, midday and
evening 1-degree intervals of air temperature values (Fig. 8). The distribution of all analysed
frequencies of occurrence of air temperature series is close to normal, particularly in summer and
autumn (see values of skewness and kurtosis rarely exceeding ±1).

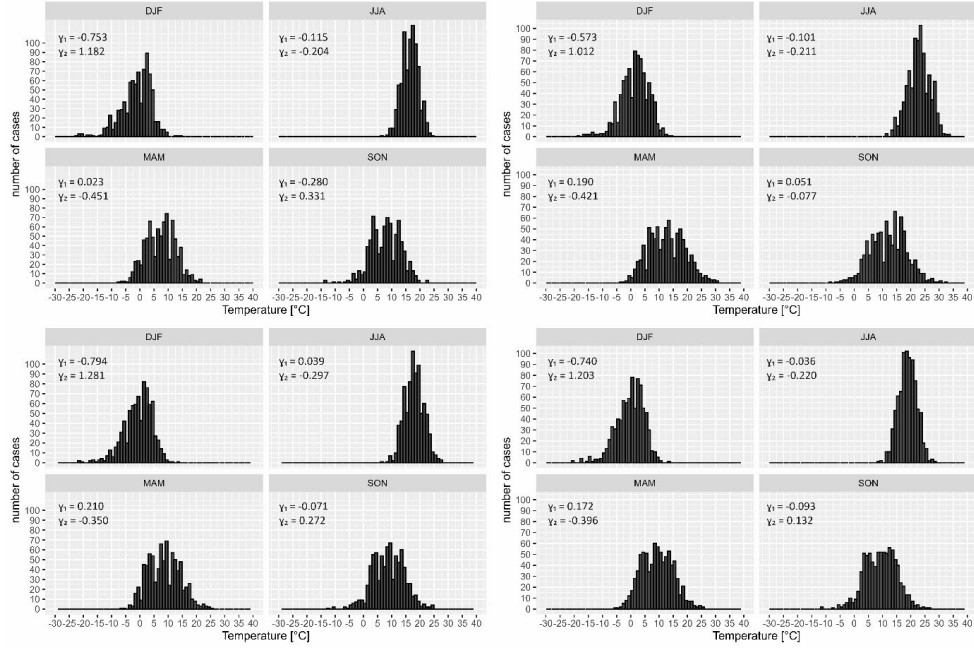


Fig. 8. Frequency of seasonal (DJF, MAM, etc.) occurrence of morning (upper left), midday (upper right), evening (lower left),
and mean daily (lower right) air temperature in Wrocław from 1773 to 1781 stratified according to 1-degree intervals
Key: $\gamma_1$ – skewness, $\gamma_2$ - kurtosis

To check whether the period 1773–81 was warm or cool, average monthly temperature values
were compared against the nine-year periods starting 100, 200 and 240 years later (Fig. 9a). Only the
most recent period (2013–21) was warmer (by, on average, 2.1 °C) than the analysed historical period.
In the annual cycle, the differences between these two periods were greatest in summer (2.8 °C) and
winter (2.5 °C) and smallest in spring (1.4 °C) (Fig. 9b). The two other comparison periods, i.e. 1873–
81 and 1973–81, were colder (by, on average 1.2 and 0.3 °C, respectively). The only month that was
colder in the historical period than in the other comparison periods was January (Fig. 9).




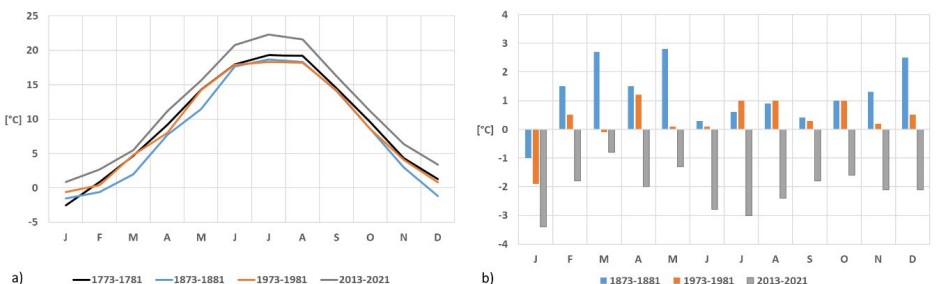


Fig. 9. Comparison of the annual cycle of air temperature in Wrocław (SW Poland) in historical time (1773–81) against other
more modern periods (a) and their differences (b)
Key: Means for the compared periods were calculated based on data from three measurement times: 1873–81 (unknown hours), 1973–81
and 2013–21 (6, 12, and 21). Air temperatures from the modern period were subtracted from those from the historical period.

4.    Discussion and Conclusions
The discovery of a new series of meteorological observations made in Wrocław (Poland, central
Europe) clearly demonstrates that, even today, there is a need to visit archives and libraries to search
for early instrumental meteorological observations. The significance of the described and analysed
series of measurements extends not only to a better understanding of the region's climate during the
period under study, but also to the future possible extension of the currently available series of
continuous measurements in Wrocław (which currently starts at 1791) back to 1773. This is possible
because meteorological data for Żagań near Wrocław are available for the period 1781–92 (see
Przybylak et al., 2014; Pappert et al., 2021). (This station was called Sagan [in Latin] within the
Mannheim network of stations established for Europe and North America by the Palatine
Meteorological Society in 1780.) This means that, for two years (1791–92), parallel meteorological
observations were conducted in Wrocław and Żagań. Przybylak et al. (2014) used mean daily air
temperature data from these years to calculate correlation coefficients (r), which were very high,
ranging from 0.91 in summer to 0.97 in spring and autumn. After extending it to 1773, the air
temperature series from Wrocław will be the longest in Poland; it will be six years longer than the
currently longest series, which is for Warsaw (Lorenc, 2000) and extends back to 1779.

For the period examined in this article, therefore, no other air temperature data exist for

south-west Poland, making a direct comparison impossible. However, it is possible to compare them
against the closest series of measured temperatures available for the study period for Berlin and
Prague (Fig. 10). Mean seasonal and annual air temperatures from the period 1961–90 at both stations
correlate strongly with those from Wrocław, e.g. correlation coefficients exceed 0.9, except for
summer temperature in Berlin (r=0.85). For the period 1773–81, there are also data from Gdańsk,
located on the Baltic coast (Filipiak and Miętus, 2010; Przybylak, 2010; Filipiak et al., 2019); however,



the correlation with the Wrocław data is clearly weaker, particularly in summer (r=0.76) and in autumn
(r=0.8), and these data were therefore omitted from the comparison. Additionally, as shown in Fig. 11,
the pattern of temperature change between the historical and contemporary periods, according to the
ModE-RA paleo-reanalysis (Valler et al., 2024), differed between northern Poland and south-western
Poland, particularly in winter and spring (as also evident in the annual values). Temperature data from
this paleo-reanalysis for Wrocław for the period 1773–81 were compared against the instrumental
observations analysed in this paper (see Fig. 10a). It can be seen that all monthly means from October
to May reconstructed by the ModE-RA paleo-reanalysis were colder than the measured values, with a
maximum difference in November (2 °C) and December (1.9 °C). On the other hand, the summer
months of July and August were, respectively, 1.7 °C and 1.3 °C warmer than observations. The
question arises as to the reason for these discrepancies. They are certainly attributable both to the
imperfection of the paleo-reanalysis for Poland and to possible biases in the air temperature series
analysed here. It appears that the primary weakness of the temperatures reconstructed by the ModE-
RA paleo-reanalysis for Poland, including the Wrocław area, lies in its limited use of proxy data sources
from Poland in the temperature reconstruction. Between 1773 and 1778, there were only three
records for the warm half-year (two documentary sources from central and south-western Poland and
one instrumental series from Gdańsk) and four for the cold half-year (in addition to the
aforementioned records, information was available on sea-ice coverage in the Baltic Sea). Additionally,
instrumental observations from Warsaw were assimilated into the model between 1779 and 1781.
Therefore, caution must be exercised when taking into account the reconstructed temperatures in the
ModE-RA paleo-reanalysis for Poland (including Wrocław). The presented instrumental series, despite
entailing some uncertainties should, when assimilated into the model used for reconstructing
temperature, help improve the quality of this paleo-reanalysis.

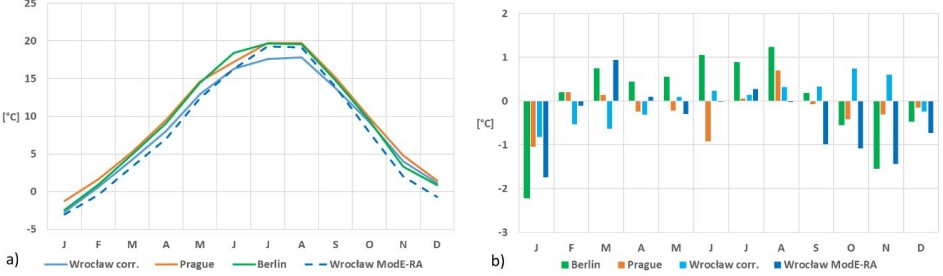


Fig. 10. Comparison of the annual cycle of air temperature in Wrocław in historical time (1773–81) against other available
series in central Europe (a), and their differences in relation to the 1961–90 reference period (b). Key: Air temperatures from
the reference period were subtracted from those of the historical period.



Fig. 11. Seasonal and annual mean air temperature anomalies in the European-Atlantic region in the period 1783–81 relative to 1961–90 according to the ModE-RA paleo-reanalysis (ClimeApp – Valler et al., 2024). Black, green, and purple dots mark Wrocław, Berlin, and Prague, respectively.



Taking into account the above-mentioned reservations regarding possible inaccuracies in the
reconstruction of air temperature in Wrocław taken from ModE-RA, as well as possible biases in the
analysed instrumental series related to, among others, the lack of knowledge about the exact exposure
of thermometers or the hours of measurement, we also present their comparison against
temperatures from the closest stations, i.e. Berlin and Prague (Fig. 10). Data from these stations for
the study period (1773–81) were assimilated into the model used for European temperature
reconstructions by the ModE-RA paleo-reanalysis. In line with expectations, Wrocław is colder than
Berlin and Prague in both projections of the annual cycle (corrected instrumental series and ModE-
RA), whereas the monthly means in Prague and Berlin are very similar (Fig. 10a).
Figure 10b shows the change in air temperature in central Europe (western Czech Republic,
south-western Poland, eastern Germany), represented by stations in Prague, Wrocław and Berlin,
between the historical period (1773–81) and the contemporary period (1961–90). The reference
period (1961–90) was the same as that used in the ModE-RA paleo-reanalysis, which allows for direct
comparison of results. The following pattern of temperature changes is observed: generally (with some
exceptions), in the historical period, the warm half-year was warmer, whereas the cold half-year
(October to February) was colder. More consistent relations exist when seasonal and mean data are
taken into account. According to the presented instrumental data, the mean annual air temperature
between the two periods changed very little. During the historical period, the temperature in Berlin
was only 0.1 °C higher than it is now; in Wrocław, there was no change; but Prague was 0.2 °C colder.
The temperature changes between the two studied periods correspond most closely for winter and
summer. In Berlin, Wrocław and Prague, the winter mean temperature was colder in historical times
than at present (by 0.2, 0.8, and 0.3 °C, respectively). On the other hand, the summer mean
temperature in the historical period was 0.8 °C higher than at present in Berlin and 0.2 °C higher in
Wrocław, while Prague was slightly (0.1 °C) colder. Very good coherence between seasonal means
(except autumn) and the annual mean is seen between the Prague and Wrocław stations (differences
range from 0.2 to 0.5 °C). In autumn in Wrocław, the temperature was higher than it is currently,
particularly in October and November, whereas it was lower at the other two stations – and for
Wrocław temperatures extracted from ModE-RA (see Fig. 10b).
Generally, the results presented here for the temperature change from the historical period
(1773–81) to the present period (1961–90) based on instrumental measurements are in good
agreement with the change derived from reconstructions available in the ModE-RA paleo-reanalysis
(compare Figs. 10 b and 11). When data from Wrocław are compared, there is quite a large discrepancy
between results for autumn months and March (Fig. 10b). In other months, the differences are small.
We should assume that the instrumental data are of better quality than the reconstructed temperature
presented in the ModE-RA paleo-reanalysis. Therefore, it seems reasonable to state that, assimilating



the instrumental series of air temperature presented here will improve the temperature
reconstructions available from the ModE-RA paleo-reanalysis for the study's historical period. We hope
that such an adjustment can be made in any subsequent version of the ModE-RA paleo-reanalysis,
because not only has this series been discovered, but so to have others for other historical periods
(see, e.g., Przybylak et al., 2025).
Summary of key results:
1.   The analysis of the newly discovered unique observational series from Wrocław for the end of

the 18th century presented herein is very valuable due to the lack of any other instrumental

data for the years 1773–78 for southern and central Poland (the Warsaw series starts at 1779,

Lorenc, 2000).

2.   The presented comparative analysis of the newest Wrocław temperature series against the

available temperature reconstruction from the ModE-RA paleo-reanalysis and with

instrumental data from neighbouring areas (Berlin and Prague) showed that the Wrocław

series is reliable and allows for a credible description of the climate of SW Poland in the studied

time.

3.   The reliable air temperature data obtained for Wrocław for the years 1773–81 will be the

subject of our future analysis in a separate paper aiming to extend the series of regular

observations of air temperature (which started here in 1791) back to the year 1773. This is

possible because, for the period 1781–92, air temperature series are available for Żagań near

Wrocław (Przybylak et al., 2014; Pappert et al., 2021).

4.   The air temperature in Wrocław during the period 1773–81 was 1.2 °C and 0.3 °C warmer than

in the respective equivalent periods 100 and 200 years later (i.e., 1873–81 and 1973–81) but

2.1 °C colder than in the most recent period (2013–21). The greatest increase in temperature

between the studied period and the latter period occurred in summer (2.8 °C) and the smallest

in spring (1.4 °C) (Table S2, Fig. 9).

5.   On average, in line with expectations, the warmest month was July (19.3 °C) and the coldest

was January (-2.5 °C) (Table S2, Fig. 4).

6.   The highest monthly temperature (22.0 °C) was calculated for August 1781 and the lowest

(-9.3 °C) for January 1776 (Table 1). On the other hand, the highest temperature (33.9 °C) was

recorded at midday on July 4, 1781 and the lowest (-22.8 °C) in the morning of January 27,

1776.

7.   In the study period, the mean seasonal and annual air temperatures do not show any trend.

The mean annual temperature is close to the mean temperature for spring and autumn (Table

S2, Fig. 5).



8.  Historical-to-present temperature changes in mean seasonal values (except autumn) for the
Prague station cohere very well with those for Wrocław (differences range from 0.2 to 0.5 °C).
The temperature agreement between Wrocław and Berlin is, overall, slightly weaker;
nevertheless, for some months, it is, in fact, better (e.g., May, June and September; see Fig.
10b). Additionally, the agreement between mean annual temperatures is closer, with the
difference being only 0.1 °C.


**Author contributions**. Study design by RP. Source analysis by PO, AP and RP. Data collection and
selection by AP and RP. Data curation by AP and RP. Literature review by RP. Statistical analysis and
visualisation by AP and RP. Interpretation of results by RP and AP. Preparation of the manuscript by RP
with contributions from all co-authors.

**Conflicts of interest**. The authors declare that they have no conflict of interest.

**Acknowledgements**
The work was supported by the National Science Centre, Poland, project No. 2020/37/B/ST10/00710.
The authors thank Ms Zuzanna Sobierajska for digitising the original data.
**Data Availability Statement**
Datasets for this research were derived from the following public domain resources:
1)  Repository for Open Data (RepOD), Nicolaus Copernicus University Centre for Climate Change
Research collection, https://repod.icm.edu.pl/dataverse/ncu-cccr, as cited in Przybylak and
Pospieszyńska (2025)
2)  Wrocław – 1873-81 and 1961-90 – Bryś and Bryś 2010b, https://doi.org/10.12775/bgeo-
2010-0007; 1973-81 and  2013-21 – Public database of Institute of Meteorology and Water
Management – National Research Institute (IMGW-PIB)
3)  Berlin – 1773-81 – https://doi.pangaea.de/10.1594/PANGAEA.870862, Berlin Tempelhof
1961-90,
https://opendata.dwd.de/climate_environment/CDC/observations_germany/climate/monthl
y/kl/historical/.
4)  Prague – 1773-81 – Meteorologická pozorování v Praze-Klementinu 1775-1900, 1976, HMU
Praha and Marciniak K., Kożuchowski K., 1990, Aneks. In: Kożuchowski K. (ed.), Materiały do
poznania historii klimatu w okresie obserwacji instrumentalnych, wyd. Uniw. Łódzkiego, Łódź:
302-452, updated to 1990.
5)  ModE-RA – ClimeApp (Valler et al., 2024)






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
