# Peer review of "Air temperature changes in Wrocław (SW Poland) in 1773-81 based on a newly discovered series of meteorological measurements Rajmund PRZYBYLAK1,3 (ORCID: 0000-0003-4101-6116) Aleksandra POSPIESZYŃSKA1,3 (0000-0003-2532-7168) Piotr OLIŃSKI2,3 (ORCID: 0000-0003-1428-080"

_EGUsphere, 2025_

## Referee Comment (RC2)

Early meteorological measurements and observations are valuable in many respects. Long instrumental measurements serve to analyze climate variability in general, climate variability with lower greenhouse forcing than currently, and to put climate change into perspective. Furthermore, shorter instrumental measurements and observations are used for climate field reconstructions, such as in data assimilation. Meteorological measurements date back to the 17th century and, compared to proxy data, offer higher temporal resolution and, in general, lower uncertainty. Accordingly, temperature and other meteorological observations/measurements are of importance to the scientific community and within the scope of Climate of the Past. Although I have numerous concerns, I would like to see this data/paper published after major revisions.

Main points:

I tried to separate the points, but in general, they overlap to some extent, as my main point is that the other measurements and observations should also be published in this paper.

1) More historical research:
   I would have liked to read more about J.E. Scheibel. As little metadata about the measurements is known, it might help to gain more information, for instance, a hint on the measurement times. Furthermore, more in-depth research in general. Maybe the obvious hygrometer reading could indicate whether he obtained his measurement devices from a meteorological network. This would also give an idea or an educated guess of which other measurement devices he used. The knowledge of metadata is important for the homogenization of measurements.

2) Early measurement errors.
   Inhomogeneities are expected in early meteorological measurement series. I don't want to list references here, but there are numerous papers on the topic. Accordingly, I would have expected more in-depth research here, as even in the absence of metadata, it is possible to validate the measurements and observations to a certain extent (see the next point).

3) Inclusion of other meteorological measurements and observations:
   I do not understand why the other records are not published (see Fig. 2). This would also improve the paper's quality, as you could compare the different measurements and observations to determine whether they are climatologically sound. For instance, checking the diurnal temperature range on clear, non-precipitation days could provide insights into measurement times, where the measurement device is mounted, whether it is shielded, etc. Or by comparing air pressure with temperature.

4) Different structure:
   For me, the current state of the paper does not convince me that one should trust the data; put differently, a validation is missing or appears later in the text.

Accordingly, a better approach would be to validate the measurements using available possibilities (contemporary and modern measurements). After that, the time series and others could be presented and discussed.

Minor revisions:

Title:

[Line 1; Title]: I personally would shorten and focus on the relevant information: "Instrumental temperature measurements in Wroclaw (southwestern Poland) from 1773 to 1781". Especially the *"based on a newly discovered series of meteorological measurements"* should be omitted, as it is not the paper's topic. What should be revised is the phrase *"newly discovered",* where *"newly"* is just redundant. Maybe better: "recently discovered".

Abstract:

In general, the readability of the abstract is low, with most sentences beginning with "The" and a noun. There could and should be more variety in writing the sentences, as the abstract is the most important part of the paper.

[Line 15]: Again, rewrite *"newly discovered".*

[Line 16]: *"... and its importance for climate analysis."* Either state why it is important or delete this phrase. (For instance, it is important for analyzing long-term climate variability or as additional data in data assimilation reconstructions, and more.)

[Line 19]: If the other meteorological measurements are not the subject of the paper, leave the information about those in the abstract.

[Line 24 to 27]: I don't think that the three sentences are relevant enough to be written in the abstract, especially the first two sentences.

[Line 15 to 31]: I, as a reader, would like to know if I can trust the data. Thus, add two sentences about validation with other time series (Berlin or Prague records).

Introduction:

[Line 44]: *"Its main goal is to find, in libraries and archives, historical sources (e.g., written records such as sagas, chronicles, maps and early instrumental measurements) that contain meteorological measurements or weather descriptions and then to record them, preferably in digital form."*

I somewhat disagree here. Either write "One of its main goals is to find, ..." or include the other very important part of data rescue, which stands for deciphering, interpreting, and transforming early instrumental readings into modern units.

[Line 43 to 50]: Somewhere here, I would expect to get information on why it should be important to be interested in early instrumental measurements. In short, why is the work important?

Area, Data, and Methods

[Line 112, Fig.1]: Maybe a bit picky, but on the historical map, I would change the red dot to a rectangle over the block to mark where the measurements were performed. This is because then I could zoom in and could get an impression of, if measured outside, possible measurement errors (are buildings reflecting sunlight, and so on...).

[Line 123]: Keep one date format, i.e., in contrast to line 101.

[Line 124]: *"The quality of this first issue of the newspaper is poor, ..."* Reading this, I would expect a clarification of how poor the quality is. Did you have difficulties reading the measurements? If so, how did you address this issue? In general, it would be interesting to read a bit more about the digitization process (readability of the newspaper copies, strategies to avoid unreasonable values, and more). A few lines later, there are a few sentences about it, but it could be more extensive.

[Line 127; Fig. 2]: Add an explanation for the columns in English. For instance, the first column shows the month, the second column provides brief weather comments, the third column shows precipitation, ... Also, what "Mo", "Mi", and "Ab" mean as referring to the times of the day, morning, midday, and evening.

[Line 138; Fig. 3]: Either skip this Figure as it is not that important information to have a Figure for it, or add a second zoomed figure, where one can also see the flag. As it is now, it proves only that one sees the tower, though, at least I can't see a flag.

[Line 146]: Do you know more about the hygrometer measurement device? Name of the instrument, scale, dimension of the scale, and whether it measures relative or absolute humidity. Also, explain (or cite) why you know that the measurement device was produced (invented?) by Prof. Johann Heinrich Lambert. Furthermore, how do we know the dimensions of temperature and pressure measurements? Is it a hypothesis by you, or is it explicitly stated somewhere (at least it is not in Fig. 3)?

[Line 142 to 149]: Already mentioned most things in the former point, but I think the following should be made clearer here: What do we know about the instruments and the measurement itself? From where do we know that? And state explicitly what we don't know. For instance, the sentence *"No further details about the exposition or precise*

*location of the instrument installation are available.”* could imply that everything else is known about measurement devices, measurement procedures, and so on.

[Line 152]: There are reasons for choosing this weighted mean formula. Add these reasons.

Results:

[Line 186]: *“The newly discovered series of meteorological measurements for Wrocław for the historical period 1773–81 is quite long (9 years).”*

Not really sure what this sentence adds as additional information. And if it is long, compared to what?

[Line 190]: *“…, we initiate an analysis of climate conditions, focusing primarily on air temperature…”*

The analysis is only on temperature. Thus, you are analyzing temperature rather than climate conditions.

[Line 186 to 192]: In general, this content has either already been introduced or fits better in the introduction or in the Data & Methods section. In any case, it does not introduce the result section.

[Line 193, subsection title]: “Yearly, seasonal, and monthly resolution” might be more appropriate, as a yearly and seasonal analysis is also made in this subsection.

[Line 196]: *“On average, in line with expectations, the warmest month was July (19.3 °C) and the coldest was January (-2.5 °C). Only slightly colder than July was August (19.2 °C). On the other hand, the other two winter months (December and February) were much warmer than January, with even positive temperatures of 1.3 and 0.9 °C, respectively (Table S2, Fig. 4).”*

Especially the first sentence is trivial, so I would rephrase these sentences in something like:

Figure 4 shows the expected unimodal annual cycle of monthly means, with July (19.3 °C) and August (19.2 °C) as the warmest months and January (-2.5 °C) as the coldest. Compared to January, December (1.3 °C) and February (0.9 °C) appear warmer.

[Line 207]: Inside the table, instead of “MEAN DAILY” write “DAILY MEAN”.

[Line 222]: *“The annual cycle based on mean monthly temperature values is clearly better approximated by the temperature observed in evening hours, particularly in the cold half-year, when the differences are about 0.4 °C (see Table S2, Fig. 4).”*

This statement is kind of obvious and can be neglected: the evening temperature better represents the mean temperature than the morning and midday temperatures, which

usually better represent the minimum and maximum temperatures. Also, your formula to calculate the mean temperature has a higher weight (two times the evening temperature) than the others.

[Line 225]: *"The air temperature was greater in evening hours than morning hours, especially in the warm half-year (see Table S2 and Fig. 4)."*

Also, obvious and can be neglected.

[Line 228]: *"The absolute range between mean monthly values measured at midday and morning hours reached 36.8 °C."*

This sentence would need more context, because I don't understand what the benefit of this information is.

[Line 231]: *"On the other hand, the highest single temperature measurement (33.9 °C) was recorded for midday of July 4, 1781 and the lowest (-22.8 °C) in the morning of January 27, 1776."*

Might fit better in the "Daily and sub-daily resolution" section.

[Line 233]: *"The preliminary analysis of the values revealed that data from midday can be treated roughly as the maximum observed value during the day. On the other hand, the morning observation seems to represent approximately the minimum temperature value for the day."*

Again, obvious and can be neglected. Because of these sentences, I would wish for a systematic analysis before the results are presented. Including, among others, for instance, a comparison with modern measurements, where one can see what one could expect (annual cycle, etc.). This would also help spot certain measurement errors when comparing, for instance, the diurnal temperature range, which could indicate an unshielded temperature instrument.

[Line 240]: This hypothesis could be checked easily. Just calculate the diurnal temperature range for the corresponding temperatures of a contemporary temperature record. If there is a significant gap between your new temperature measurements and the compared data in the diurnal temperature range, then there might be an inhomogeneity in the data.

[Figure 6]. Although the time resolutions differ (daily and monthly), Figures 4 and 6 show the same result. Thus, the conclusions of these Figures are basically similar. Skip one.

[Figure 7]: Label the y-axis differently, "Morning temperature" and accordingly the others.

[Table 2]: Reconsider this table, as most of the information in this table is already present in Fig. 7. (Higher interquartile range indicates a higher standard deviation and vice versa.)

[Figure 9 b]: It would be more reasonable to calculate the monthly mean temperature of a different time period minus the monthly mean temperature of 1773-81. Seeing negative anomalies for the most recent temperature time span is odd.

[Line 349 to361] Neglect these sentences and leave it with that ModE-RA and the temperature record show differences. The reason for that is that the conclusions drawn here are not sound and are as follows: only a small number of observations (instrumental and proxy data) are near Wroclaw, so one cannot trust the results of ModE-RA and should trust the Wroclaw records. The Ensemble Kalman filter-based data assimilation approach also includes a background covariance matrix, which is calculated from a climate model. Saying that, observations with a lower uncertainty, a high correlation (covariance matrix), and less distance than the length scale parameter (in ModE-RA for temperature in between 1500 and 3000 km) are also assimilated in this grid point. In other words, also measurements from Germany, Austria (Vienna, Kremsmünster, etc…) are assimilated in this specific grid point, although further away. However, they also published a so-called feedback archive, which allows checking which observations "suggest" which value for this grid point. That said, sure, new instrumental measurements are valuable for data assimilation, but the arguments here are not valid.

[Line 398]: Maybe a bit inconsistent here, as before it was said that the results are different between ModE-RA and the Wroclaw temperature record.

[Line 406]: *"We hope that such an adjustment can be made in any subsequent version of the ModE-RA paleo-reanalysis, because not only has this series been discovered, but so to have others for other historical periods."*

Not sure if the meaning makes sense.

[Line 410]: Write a more comprehensive summary without having a list of points. Also, it should include an outlook for publishing the other meteorological measurements (if not included in this paper; see major concerns of this comment)